# Molecular Basis of Dipeptide Recognition in *Drosophila melanogaster* Angiotensin I-Converting Enzyme Homologue, AnCE

**DOI:** 10.3390/biom15040591

**Published:** 2025-04-16

**Authors:** Joanna Żukowska, Kyle S. Gregory, Adam Robinson, R. Elwyn Isaac, K. Ravi Acharya

**Affiliations:** 1Department of Life Sciences, University of Bath, Claverton Down, Bath BA2 7AY, UK; jz2424@bath.ac.uk (J.Ż.); kg540@bath.ac.uk (K.S.G.); adam.robinson@bristol.ac.uk (A.R.); 2Institute of Integrative and Comparative Biology, Faculty of Biological Sciences, University of Leeds, Leeds LS2 9JT, UK; r.e.isaac@leeds.ac.uk

**Keywords:** angiotensin I-converting enzyme homologue, X-ray crystallography, metalloprotease, dipeptide, crystal structure, inhibitor design, peptide binding, enzyme mechanism, domain selectivity

## Abstract

Human angiotensin-I-converting enzyme (ACE) is involved in vasoregulation, inflammation, and neurodegenerative disorders. The enzyme is formed of two domains; the C-domain (cACE) is primarily involved in blood pressure regulation, whereas the N-domain (nACE) is strongly linked to fibrosis; hence, designing domain-specific inhibitors could make a difference between treating one condition without having a negative effect on another. AnCE (a close homologue of ACE) is derived from *Drosophila melanogaster* and has a high similarity specifically to cACE. Due to high similarity and ease of crystallisation, AnCE has been chosen as a model protein for ACE studies and for the design of ACE inhibitors. In this study, enzyme kinetic assays and X-ray crystallography techniques revealed the significance of using dipeptides as selective inhibitors for AnCE and how this knowledge could be applied to cACE and nACE. All the dipeptides tested in this study were shown to bind AnCE in two distinct locations, i.e., the non-prime and prime subsites. It was found that a hydrophobic residue at the S1 and S1′ subsites, with a tryptophan at the S2 and S2′ subsites, showed highest affinity towards AnCE. It was also observed that a key pocket within the S2′ subsite had a major influence on the binding orientation within the prime subsites and could potentially explain ACE’s dipeptidyl carboxypeptidase activity. Importantly these dipeptides are found in functional foods, making them potentially available from diets. Knowledge of the dipeptide binding presented here could aid in the development of ACE domain-specific inhibitors.

## 1. Introduction

Angiotensin-I-converting enzyme (ACE, EC 3.4.15.1) is a promiscuous zinc metallopeptidase with a number of substrates, including the vasoregulatory peptides angiotensin I and bradykinin, the anti-fibrotic agent N-acetyl-SDKP, the neurotransmitter substance P, and amyloid-β peptide. One of the main functions of ACE is the degradation of angiotensin I to produce the vasoconstrictor angiotensin II; due to this, ACE has become a therapeutic target for hypertension, congestive heart failure, myocardial infarction, renal disease, and diabetic nephropathy. ACE has two forms: a somatic form, which is 150–180 kDa in size, and a smaller germinal isoform expressed in the testes, which is 100–110 kDa in size [1,2,3,4,5,6].

Human somatic ACE has two homologous N- and C- domains (known as nACE and cACE) that display subtle differences in substrate specificity and inhibitor binding. These domains share approximately ~60% overall amino acid sequence identity and about 89% identity when comparing the active sites. In this context, the high resolution 3-dimensional structures of the two domains elucidated previously by us are invaluable for the design of next generation domain-selective ACE inhibitors [7,8]. While there are many ACE inhibitors in the market that block both domains, there are no clinically available drugs that selectively inhibit cACE (the dominant domain responsible for hypertension) or nACE (predominantly involved in fibrosis and inflammation in the heart, kidney, and lungs) [9]. Therefore, we have been engaged in deepening our structural understanding of ACE and ACE homologues in complex with inhibitors to aid in designing ‘domain-selective inhibitors’ of ACE.

The action of certain diet derived peptides as ACE inhibitors is well established, and there is an increasing amount of evidence to suggest that much smaller dipeptide fragments can also have this effect [10,11,12,13]. With the potential bioavailability of such compounds, these developments indicate that research into the inhibitory actions of dipeptides could be an important area to pursue. These dipeptides could be a product of digestion and could have an important role in the in vivo regulation of ACE activity [14]. A detailed study conducted by Lunow et al. reported that the dipeptides IW and AW were able to inhibit ACE activity with IC_50_ values of 0.7 µM and 20 µM, respectively [10]. In a further study, they reported that some dipeptides displayed selective inhibition of the two domains of ACE [14].

In the report presented by Lunow et al. [14] dipeptides with a C-terminal tryptophan residue were implicated in selective inhibition of cACE. It has been suggested that this is a result of the increased hydrophobicity of the S2′ pocket in cACE compared to nACE, a feature that has previously been exploited in the development of the cACE-selective inhibitors RXPA380 [15] and kAW ((5S)-5-[(N-benzoyl)amino]-4-oxo-6-phenylhexanoyl-L-tryptophan) [1,16,17,18,19]. Similarly, an aliphatic N-terminal residue improved cACE inhibition, with the dipeptides IW and VW being better inhibitors than YW and RW. Conversely, the data collected indicated that a C-terminal arginine residue might be associated with N-domain selectivity [14].

Here, we present the use of the *Drosophila melanogaster* ACE homologue, AnCE (which, in particular, shares enzymatic properties with cACE), as a model to study the structural basis of ACE inhibition. As well as having high similarity to cACE, AnCE can be easily produced in large quantities (~20 mg from 1 litre) and readily crystallises compared to ACE, making it an excellent model protein. We report the crystal structures of AnCE in complex with RW, WR, VW, IW, and YW, and determine their binding affinity through kinetic characterisation. This information could be built upon in the design of future domain-specific inhibitors of human ACE and could provide insight into how other related inhibitory peptides may occupy the active site of ACE.

## 2. Materials and Methods

### 2.1. AnCE Expression and Purification

Expression of AnCE (vector: pPiC9, cells: *P. pastoris* GS115) was carried out by inoculating a starter culture containing 25 mL of BMGY (1% yeast extract, 2% tryptone, 1% ammonium sulphate, 0.34% yeast nitrogen base, 1% glycerol, 17.4 g L^−1^ K_2_HPO_4_, and 134.1 g L^−1^ KH_2_PO_4_ pH 6.0) media in a 250 mL baffled flask. The culture was incubated overnight at 30 °C and 250 rpm. A total of 5 mL of the overnight culture was used to inoculate 500 mL of BMGY media in 2 L baffled flask, and the culture was incubated overnight at 30 °C and 250 rpm. The expression culture was harvested at 6000× *g* for 15 min at room temperature. The supernatant was discarded, and the cell pellet was resuspended in BMMY (1% yeast extract, 2% tryptone, 1% (NH_4_)_2_SO_4_, 0.34% yeast nitrogen base, 0.5% methanol, 17.4 g L^−1^ K_2_HPO_4_, and 134.1 g L^−1^ KH_2_PO_4_ pH 6.0) media at a volume of 1/5 of the starter volume and transferred to a 2 L baffled flask. The culture was induced for 48 h at 30 °C and 250 rpm with a methanol feed after 24 h to maintain the methanol concentration at 0.5%. After 48 h of induction, the culture was harvested at 14,000× *g* for 20 min at 19 °C. The supernatant was collected, and the cell pellet was discarded. The supernatant was buffer equilibrated to 20 mM Tris and 1.5 M (NH_4_)_2_SO_4_. After buffer equilibration, the supernatant showed some precipitation, this was removed by spinning at 14,000× *g* for 20 min, followed by filtration through 0.45 µm filter. The supernatant was loaded onto a 5 mL Phenyl HP column (Cytiva, Washington, DC, USA) at 5 mL/minute pre-equilibrated with buffer A (20 mM Tris, 1.5 M (NH_4_)_2_SO_4_ pH8.0). Following loading, the column was washed with buffer A until UV returned to baseline. Bound AnCE was eluted with 70% of buffer B (20 mM Tris pH 8.0). Fractions were analysed using SDS-PAGE. Fractions containing protein were purified further using a 16/60 Superdex 200 pg column (Cytiva, Washington, DC, USA) with buffer C (150 mM NaCl, 20 mM Tris pH 8.0), and the flow rate was set to 1.5 mL/minute. Fractions were analysed using SDS-PAGE. Pure AnCE was dialysed against 1 L of 5 mM HEPES pH 7.5 and 0.1 mM PMSF. The protein was stored at −80 °C.

### 2.2. Crystallisation, X-Ray Diffraction Data Collection, and Structure Determination

The five dipeptides were all purchased from GL Biochem (Shanghai) Ltd. (Shanghai, China). Co-crystallisation experiments for AnCE with the dipeptides VW, YW, RW, and WR were performed using the hanging drop vapour diffusion technique. Immediately prior to crystallisation, AnCE at a concentration of 17 mg∙mL^−1^ was incubated with 20 mM of each dipeptide in a 1:4 ratio of dipeptide to protein at room temperature for approximately 30 min. The protein/dipeptide mixtures were mixed with reservoir solution (1.2 M sodium citrate, 0.1 M HEPES pH 7.5) at a 1:1 ratio for VW and a 2:1 ratio for RW, WR, and YW to form a 2 µL drop. Each drop was suspended on a coverslip over 500 µL of the reservoir solution (1.2 M sodium citrate, 0.1 M HEPES pH 7.5). The crystallisation experiments were incubated at 21 °C. AnCE with IW dipeptide structure was obtained through a soaking experiment, by adding 5 µL of IW at 20 mM directly to the AnCE native drop. This caused majority of the crystals to dissolve, and the remaining crystals were fished over 10 min (roughly one crystal per minute). Native AnCE crystals were set up at a concentration of 17 mg∙mL^−1^ at a 2:1 ratio of protein to condition. The rest of the steps were the same as for the co-crystals.

Previous experiments with AnCE have shown that the high sodium citrate concentration (1.2 M) used during crystallisation provides sufficient cryo-protection, negating the need for additional cryo-protectants [20]. Crystals were mounted directly into cryoloops (Molecular Dimensions Ltd., Newmarket, UK) and rapidly cryo-cooled in liquid nitrogen.

All diffraction data were collected at Diamond Light Source (Didcot, UK) using beamline I04. Data were captured using an Eiger2 XE 16M detector (Dectris, Baden-Daettwil, Switzerland) at a wavelength of 0.95 Å, and all diffraction experiments were performed at 100 K. A total of 3600 images were taken at 0.1° of oscillation with a dose of 14 MGy for VW, 13 MGy for RW and WR, and 10 MGy for IW and YW.

The IW and YW datasets collected were processed using the XIA2 DIALS pipeline [21] at Diamond Light Source. VW, RW, and WR raw images were indexed and integrated using DIALS. All further data processing was carried out using the CCP4i2 suite, and data reduction was performed with AIMLESS [22,23]. Initial phases were calculated by molecular replacement using PHASER [24] with the native AnCE structure, PDB code 5A2R [25]. Following production of the initial models, iterative rounds of model building in COOT [26] and refinement using REFMAC [27,28] were performed. The dipeptides, zinc ion, sugars, and waters were added based on the dFo-mFc Fourier difference map.

MOLPROBITY [29] was used to validate the quality of the final structures. The molecular graphics program PyMOL (The PyMOL Molecular Graphics System, Version 4.6, Schrödinger, LLC, New York, NY, USA) and CCP4 mg [30] were used to prepare the figures.

### 2.3. Inhibition of AnCE Activity

Enzyme assays were performed using the internally quenched fluorogenic peptide Abz-FRK(Dnp)-P (Enzo Life Sciences Ltd., Exeter, UK) as a substrate for 25 ng of AnCE in 200 µL of 100 mM HEPES buffer pH 7.5 with 50 mM NaCl and 10 µM ZnCl_2_. The reaction was started by adding 2 µL of 5 mM Abz-FRK(Dnp)-P in dimethyl sulfoxide to the buffer in wells of a 96-well black plastic plate (Corning Life Sciences, High Wycombe, UK), and the release of fluorescence on hydrolysis of the substrate was monitored continuously at 20 °C using a FLUOstar Omega (BMG Labtech GmbH, Offenburg, Germany) with λex at 340 nm and λem set at 430 nm. For studying the effect of dipeptide inhibitors, the enzyme in the assay buffer was pre-incubated with inhibitor for 10 min prior to the addition of the substrate. IC_50_ values were determined using the non-linear regression curve-fitting software [log(inhibitor) vs. response] of GraphPad Prism 7.01 (GraphPad Software, San Diego, CA, USA). The coefficient of determination (R^2^) values for the regression model ranged from 0.9852 to 0.9629. K_i_ values were obtained using the Cheng–Prusoff equation and a substrate K_m_ of 14.24 µM, obtained using non-linear regression curve-fitting software (Michaelis–Menten) of GraphPad Prism 7.01. The R^2^ for the Km determination was 0.9787 [31].

## 3. Results and Discussion

### 3.1. Kinetic Analysis of AnCE Inhibition

The dipeptide affinity for AnCE was tested. The IC_50_ and *K*_i_ values were determined for each dipeptide, showing that the highest to lowest affinities for AnCE were as follows: IW > VW > RW > YW > WR (Table 1). This indicates that binding to AnCE favoured a hydrophobic residue at the N-terminus and a tryptophan at the C-terminus. Reversing the tryptophan position (WR) resulted in significantly lower affinity towards AnCE. A similar finding was previously reported for cACE inhibition by RW and WR [14].

### 3.2. Crystal Structures of AnCE in Complex with Dipeptides IW, VW, YW, RW, and WR

The high-resolution crystal structures of AnCE in complex with IW, VW, YW, RW, and WR were determined by molecular replacement (search mode: native AnCE structure, PDB: 5A2R [25]) in the space group *H3* with one molecule in the asymmetric unit (Table 2).

Analysis of the overall topology of each structure indicated that there was no large-scale movement of secondary structure elements compared to the native AnCE structure. This was confirmed by the calculated root mean square deviation (RMSD) values in the range from 0.214 to 0.262 Å for each structure compared to native AnCE. Key features of AnCE seen in the structures include the long substrate binding channel that extends the length of the molecule and is capped by an “N-terminal lid” formed by helices α1, α2, and α3. The active site is located at the centre of this channel, marked by the location of the catalytic zinc ion (Figure 1), which is coordinated by His367, His371, and Glu395. In all of the dipeptide-bound structures presented here, clear Fo-Fc electron density was observed in both the non-prime and prime subsites. This density was assigned to dipeptides VW, IW, YW, RW, and WR. The dipeptides were modelled to this density as appropriate (Figure 2), and their binding is discussed in detail below. The dipeptide occupying the S1 and S2 subsites will be referred to as dipeptide-1 (IW-1, VW-1, YW-1, RW-1, and WR-1) and the dipeptide occupying the S1′ and S2′ subsites as dipeptide-2 (IW-2, VW-2, YW-2, RW-2, and WR-2).

### 3.3. Binding of IW, VW, YW, and RW Within the Non-Prime Subsites

The N-terminal amino acid backbone interactions are conserved for IW-1, VW-1, YW-1, and RW-1; through water-mediated interactions, the amine is stabilised by Asp342 and the dipeptides own tryptophan, and the carbonyl is stabilised by Arg506. IW-1 isoleucine and VW-1 valine sit within the S1 subsite, forming a hydrophobic contact with Trp341 (Figure 3A). The RW-1 arginine shows significantly more interactions when compared to IW-1 isoleucine and VW-1 valine; this is likely due to a much larger charged side chain, resulting in it being able to interact with more distant residues. The arginine interacts with the S1 subsite residues mostly through water-mediated interactions, with the exception of the arginine side chain directly forming H-bonds with Tyr496 (also observed in YW) and a salt bridge with Glu124 (which has moved relative to Glu124 in native AnCE). The side chain further interacts with the following S1 residues through a series of water-mediated interactions: Ala47, Lys62, Ser339, Lys352, Ala500, and Asp501, as well as His337 from the S1′ subsite (Figure 3B). The YW-1 tyrosine forms a H-bond with Tyr496, which is not possible in IW and VW; however, it maintains the hydrophobic contact with Trp341 observed with those dipeptides (Figure 3C).

The dipeptides with a C-terminal tryptophan (IW, VW, YW, RW) show an identical conformation of binding within the non-prime subsite, with the tryptophan residue occupying the S2 subsite (Figure 3D). The tryptophan forms a π-stacking interaction with His394, hydrophobic interactions with Phe375 and Pro391, and water-mediated interactions with Tyr378, Arg386, Tyr344, and Asp342 via the indole nitrogen. There are also conserved backbone interactions, where the amide group forms H-bonds with Ala340, and the carboxyl group interacts with Arg506, Tyr507, Glu395, Ala338, zinc ion, and the second dipeptide through water-mediated interactions.

### 3.4. Binding of IW, VW, YW, and RW Within the Prime Subsites

The N-terminal backbone interactions are conserved for IW-2, VW-2, YW-2, and RW-2. The amine group forms H-bonds with Ala338 and Glu368, in addition to water-mediated interactions with the zinc ion and carboxyl group of dipeptide-1. The carbonyl group is positioned towards the S2′ subsite, thus allowing for the formation of H-bonds with His337, His497, and Tyr507.

IW-2 isoleucine occupies the S1′ subsite, forming hydrophobic contacts with Ala338 and Thr364. Due to the similarity of isoleucine and valine, the interactions within the S1′ subsite for VW-2 and IW-2 are conserved (Figure 4A). Based on this, it is likely that the difference in the dipeptide affinities towards AnCE (IW: 0.08 mM, VW: 0.17 mM) is influenced by the difference in hydrophobicity, with a more hydrophobic side chain being favoured. The RW-2 arginine side chain forms the most interactions out of all the dipeptides listed in this section. The side chain forms H-bonds with Gln266 and Thr364 and water-mediated interactions with Glu150, Asn261, Gln265, and Gln361. The arginine also forms a salt bridge with Asp360 and hydrophobic contacts with Ala338 and Thr364 (Figure 4B). In YW-2, the tyrosine side chain forms hydrophobic contacts with Ala338 and Thr364 and a H-bond with Thr364. Interestingly, YW-2 folds in a conformation that allows for a π-stacking interaction between the tyrosine and tryptophan of the dipeptide (Figure 4C). The additional difference map density around the tyrosine side chain in the S1′ subsite resulted in alternative conformations of the side chain to be tested, as well as dual conformations with the whole dipeptide flipped. The electron density remained ambiguous; for this reason, the dipeptide was left in a singular conformation. This ambiguity could be caused by lower affinity binding displayed by the dipeptide (Ki: 0.75 mM), which, in turn, could be caused by Thr364, though forming a H-bond, it could also have a repelling effect on the aromatic ring. Interestingly, in ACE, cACE has a slight preference for YW when compared to nACE, this could be caused by a threonine residue being present in nACE but not in cACE, which, in turn, is replaced by a valine.

The tryptophan binding within the S2′ subsite is conserved between IW-2, VW-2, YW-2, and RW-2, where the tryptophan side chain forms hydrophobic interactions with Phe441 and Tyr507, a π-stacking interaction with Phe511, and a H-bond with Gln266 (Figure 4D). The position of the tryptophan is most likely due to the backbone orientation being restricted by the carboxyl group sitting within the S2′ polar pocket created by residues Gln265, His337, Lys495, His497, and Tyr504. This pocket is conserved between AnCE, nACE, and cACE and has been shown to coordinate the carboxyl group of other ligands, such as RXP380 (cACE: 2OC2 [32], AnCE: 2X96 [33]), Omapatrilat (cACE: 6H5W, nACE: 6H5X [34]), and sampatrilat (cACE: 6F9T, nACE: 6F9V [35]). The anchoring of the C-terminal carboxyl group and N-terminal carbonyl positions the N-terminal amine group in close proximity to the catalytic zinc ion, potentially mimicking the cleaved products of substrates. This is further supported by previous studies, where on co-crystallisation of nACE with the antifibrotic peptide Ac-SDKP, the cleaved terminal dipeptide (KP) is observed with the N- and C-terminus anchored identically to the dipeptides presented here [36].

### 3.5. Binding of WR Within Non-Prime Subsite

Due to ambiguity in the electron density maps, WR-1 was modelled in dual conformation in the non-prime subsites. In the first conformations, the arginine faces towards the S2 subsite, and the tryptophan sits within the S1 subsite, whereas in the second conformation, the dipeptide is flipped. In the first conformation, the arginine carboxyl group coordinates the zinc ion through a water-mediated interaction, whereas in the second conformation, the dipeptide distance from the zinc ion is too far for coordination. The electron density for WR-1 makes it difficult to know exactly how to position the arginine within the density in either conformation, as the tryptophan has a clear difference map in both conformations (Figure 2E,F). However, as there is no difference map density around the arginine positions, they were modelled with some confidence, and the interactions for the residue can be described. 

The first conformation of WR-1 orientates the tryptophan towards the S1 subsite, where it forms a water-mediated interaction with Arg506 and hydrophobic contacts with Trp341, Tyr496, and Val502. The arginine backbone forms a H-bond with Ala340 and water-mediated interactions Glu395, Arg506, and Tyr507. The carboxyl group coordinates the zinc ion and WR-2 through a water-mediated interaction in the same way as IW-1, VW-1, YW-1, and RW-1. The long side chain of arginine also allows for a H-bond with Thr387 and a water-mediated interaction with Gly388 (Figure 5A).

The WR-1 tryptophan in the second conformation forms similar side chain contacts as IW-1, VW-1, YW-1, and RW-1 through a π-stacking interaction with His394, hydrophobic contacts with Phe375 and Pro391, and water-mediated interactions with Asp342, Tyr344, Tyr378, and Arg386. The carbonyl group shows a water-mediated interaction with Arg506, and the amide group forms a H-bond with Ala340. The arginine forms H-bonds with Ala340, Asp342, and Tyr496, a water-mediated interaction with Tyr344, and a hydrophobic contact with Trp341 (Figure 5B).

Based on the environment, the second conformation would be more favourable for WR-1 (as shown in the other dipeptides). This is due to higher hydrophobicity of the S2 subsite compared to the S1 subsite. Nonetheless, the electron density is clear for the carboxyl group coordinated towards the zinc ion, which indicates that the dipeptide occupies the first conformation relatively often, even if it seems like a less favourable environment (Figure 2F). The observed Fo-Fc density makes a strong case for both conformations of WR-1 (Figure 2E,F).

### 3.6. Binding of WR Within Prime Subsite

The WR-2 dipeptide was modelled in a single conformation; however, for this dipeptide, the side chain density is less continuous (Figure 2E,F). This is either due to the dipeptide’s occupancy not being at 100% or flexibility in the side chain positions. As the WR dipeptide is the weakest binder out of all the dipeptides tested (*K*_i_ = 0.84 mM) in this report, the low occupancy would not be surprising. Occupancy refinement was performed, but it did not improve the map as the occupancy was lowered.

The WR-2 tryptophan side chain interacts with Asp360 and Gln361 through water-mediated interactions and a H-bond with Thr364. The amine group coordinates the zinc ion and WR-1 (first conformation) through a water-mediated interaction, and it is also stabilised through a H-bond with Ala338. The carbonyl group forms H-bonds with His337, His497, and Tyr507 (Figure 5C).

The arginine of the WR-2 sits within the S2′ pocket and forms H-bonds with Gln266, a hydrophobic contact with Phe441 and Phe511, and a salt bridge with Asp437. This binding also causes Gln266 to move (relative to the apo structure) in order to accommodate the arginine side chain within the subsite. The electron density map for the arginine and Gln266 shows that the arginine has some flexibility within the subsite. However, upon performing alternative conformation refinement, it was found that the arginine occupies the second conformation less than 10% of the time; for this reason, only one conformation is shown in all figures. The carboxyl group is stabilised by the polar pocket made up from Gln265, His337, Lys495, His497, and Tyr504 (Figure 5D).

The change in binding conformation of WR compared to the other dipeptides appears to be highly driven by the backbone interactions, in particular the carboxyl group, which occupies the polar pocket within the S2′ subsite. It would appear that backbone binding is favoured over the side chain orientation of the dipeptide, showing that the subsites are able to accommodate a range of residues if necessary. This is further shown by the comparison of WR-2 and RW-2 binding discussed below.

### 3.7. Comparison of RW and WR Binding

RW and WR dipeptide binding was compared due to the high similarity of the peptides, the only difference being in the backbone. Interestingly, the orientation of peptides binding within the active site appears different upon comparison (Figure 6). In WR-1, there appears to be dual conformation of the dipeptide binding within the non-prime subsite, whereas with RW-1, there is clear density for a single conformation (Figure 2C). However, even when comparing similar conformations between the two dipeptides, the positioning in the active site is slightly different, with the arginine of RW-1 extending into the S1 subsite, resulting in a different position in space when compared to the WR-1 arginine. The second position of the dipeptide is also different when comparing RW-2 to WR-2. RW-2 tryptophan is positioned towards the S2′ subsite, and the arginine sits within S1′, but in WR-2, the opposite is true.

The difference in orientation between RW and WR likely favours RW binding due to the tryptophan occupying the more hydrophobic S2 and S2′ subsites. However, the dual conformation within the non-prime subsites of the WR dipeptide indicates flexibility in its orientation. Dipeptides within the prime subsites do not show this flexibility. As mentioned above, this is speculated to be due to the carboxyl group having a high affinity for the polar pocket within S2′ subsite causing the side chains to occupy less favourable environments. Additionally, this is a promiscuous enzyme that cleaves peptides at the C-terminus; hence, it needs to be able to stabilise a range of C-terminus amino acids within the prime subsite to make this possible. It is possible that within AnCE, and potentially nACE, and cACE, this conserved polar pocket is the key factor determining peptide binding within the S1′ and S2′ subsites. Interestingly, this mode of binding has been shown in the cACE and Ang II complex structure, where the C-terminus of Ang II occupies this polar pocket (PDB: 4APH [37]). The less favourable binding of WR-2 is reflected in the comparison of B-factors between WR-2 and RW-2, with WR-2 having average B-factors of 54.09 Å^2^ compared to RW-2 averaging at 40.01 Å^2^ (Table 2). This is further supported by the statistics with the WR dipeptide having the highest *K*_i_ value (0.84 mM) out of all the dipeptides tested in this study. Although, YW also has a high *K*_i_ value (0.75 mM), potentially indicating that aromatic groups are not ideal for binding within the AnCE S1′ subsite; as speculated earlier, this could be due to Thr364, which could be causing a destabilising effect on the hydrophobic rings.

### 3.8. Dipeptides Modelled in cACE

The AnCE structures in complex with dipeptides were superimposed on the apo cACE structure (PDB:1O8A [7]) to assess the possible binding within the enzyme. The overall fold between cACE and dipeptide-bound AnCE is near identical with the RMSD values of 0.79–0.82 Å (Figure 7). It is likely that all dipeptides would adopt a similar binding orientation to AnCE due to the highly conserved active sites between the two enzymes. However, there are some residue substitutions that could alter how the dipeptides interact with the active site (Table 3). The substitutions result in a slightly more polar S1 and S2 subsites but a more hydrophobic S1′ and S2′ subsites.

The changes within the S1 subsite could influence the more hydrophobic dipeptides IW-1 and VW-1, particularly due to the S1 subsite environment being less favourable for the isoleucine and valine. However, the substitution of AnCE-Thr387 to cACE-Glu403 in the S2 subsite could allow for the formation of a H-bond with the tryptophan, further stabilising it within the S2 subsite (Figure 7A,C). It is likely that these changes could be more favourable for RW-1 and WR-1 than IW-1 and VW-1 due to the changes in polarity and the larger side chains of the dipeptides. There is also an additional difference in the S1 pocket of AnCE-Tyr496 to cACE-Phe512; this change could help stabilise IW-1 and VW-1 in the cACE S1 subsite, as well as potentially having a repelling effect on RW-1 and WR-1, directing the side chains more towards the polar substitutions (AnCE-Ala47 and Gly51 to cACE-Asn66 and Asn70) (Figure 7G,I). Interestingly, WR-1 has a dual conformation in AnCE, the more polar environment of the S1 subsite in cACE could result in WR-1 undertaking a singular conformation, with arginine being solely in the S1 subsite. The substitution to Phe512 in cACE could influence the binding of YW-1, as this would remove the possibility of the H-bond within this region, and instead be replaced with potential hydrophobic interactions. Additionally, the substitution of AnCE-Ala47 and Gly51 to cACE-Asn66 and Asn70 could result in a shift of the YW-1 tyrosine towards these residues to form H-bonds, still allowing for the hydrophobic contact with Phe512 (Figure 7E).

The S1′ and S2′ subsite changes between AnCE and cACE could result in higher affinity binding of the IW-2 and VW-2 dipeptides within these subsites due to the changes from AnCE-Thr364 and Gln266 to cACE-Val380 and Thr282, respectively, resulting in more hydrophobic pockets. Interestingly, AnCE-Thr364 is also a threonine in nACE (Thr358), which likely contributes to nACE having lower affinity for IW and VW compared to cACE. The change of AnCE-Asp360 to Glu376 in cACE would most likely not affect the IW-2 and VW-2, as the distance between the dipeptides and the residue is too large (Figure 7B,D). The Asp360 change to Glu376 would likely have an effect of the YW-2 and RW-2 ligands, as the larger side chains occupying the S1′ can form interactions with Asp360. The shorter distance created by the Glu376 in cACE could lead to a stronger H-bond, resulting in tighter binding within the S1′ pocket (Figure 7F,H). These changes could also be favourable for the WR-2 mode of binding with the tryptophan facing towards the S1′ subsite due the more hydrophobic environment, though the more hydrophobic environment of the S2′ subsite would be less favourable for the arginine (Figure 7J).

Interestingly, a larger peptide (RXPA380) takes the exact same conformation in both AnCE (PDB: 2X96 [33]) and cACE (PDB: 2OC2 [32]) as the dipeptides within the prime subsites (Figure 8). The conformation of binding is different for the moiety at the S1 subsite, though the P2 phenylalanine-like structure occupies a similar orientation to the dipeptide tryptophan within the S2 subsite. This observation follows the trend of the carboxyl group being the main driving force for the binding orientation within the S1′ and S2′ subsites. Due to the phosphate group coordinating the zinc ion in the RXPA380 structures, only the P2 phenylalanine-like moiety reaches as far as the dipeptides in the same position; however, the phenylalanine forms the same π-stacking interaction with His394 as the dipeptide tryptophan, further supporting that this histidine (conserved in cACE as His410) is of importance in stabilising aromatic residues at this position (Figure 8).

Overall, it is likely that the dipeptides would bind in a similar conformation in AnCE and cACE with some potential differences in binding affinities due to small changes in the subsite’s residues.

### 3.9. Dipeptide Modelled in nACE

Similarly to cACE, the dipeptides were superimposed into the apo structure of nACE (PDB: 2C6F [8]), and the RMSD values ranged from 0.86–0.88 Å (Figure 7), indicating similar folds. The non-prime and prime subsites showed an increase in polarity in nACE when compared to AnCE, this most likely would affect binding of the dipeptides within nACE (Table 3).

Based on the residue differences in the S1 and S2 subsite (Table 3), it is likely that IW-1 and VW-1 would not bind as strongly within the subsite, as the increased polarity would not be favourable for the hydrophobic dipeptide (Figure 7A,C). When looking at WR-1, it is hard to predict if the changes within nACE compared to AnCE would favour one conformation over the other, as both the subsites could easily accommodate the arginine at either the S1 or S2 subsite. The residue substitution from AnCE-Tyr496 to nACE-Phe490 in the S1 subsite and AnCE-Phe375 to nACE-Tyr369 in the S2 subsite could favour the tryptophan within the S1 subsite and the arginine within the S2 subsite. However, the AnCE-Thr387 being substituted to nACE-Arg381 in the S2 subsite could create a repelling effect on the dipeptide arginine within the pocket, making it more favourable for the arginine to bind within the S1 subsite. Although, this repelling effect could be overcome by the dipeptide arginine being shielded by water molecules, which, in turn, could result in the arginine being stabilised within the S2 subsite by water-mediated interactions with the nACE arginine (Figure 7I). It is likely that RW-1 would bind in a similar manner in nACE as WR-1, though since in nACE RW is the less selective binder, it might display dual conformation (similar to that of WR-1 in AnCE) (Figure 7G). The substitution of AnCE-Thr387 to nACE-Arg381 could be favourable for YW-1 to bind with the tyrosine within the S2 subsite, which would allow for direct interaction with the protein arginine (Figure 7E).

The S1′ and S2′ subsites in nACE have a significant increase in polarity within the S1′ subsite, having substitutions of AnCE-Phe363 to nACE-Ser357 and the S2′ subsite AnCE-Gln266, Ser268, and Asp437 being substituted to nACE-Ser260, Glu262, and Glu431. These substitutions could cause a repelling effect on hydrophobic residues, which is further evidenced by nACE having the lowest affinity for IW and VW dipeptide (Figure 7B,D) [14]. nACE also does not have a high affinity for RW or YW; this is likely due to the changes within the S2′ pocket destabilising tryptophan binding, as the Ser260 in nACE could be too far to stabilise the tryptophan, and the changes to glutamic acids within nACE could also repel the aromatic rings of the tryptophan (Figure 7F,H). The opposite can be said for WR-2, as the arginine can maintain the interaction it would form with Gln266 in AnCE with the nACE Ser260 residue. The WR-2 arginine also extends past the interactions observed with the tryptophan and could make contact with nACE Glu262 and Glu432 (Ser268 and Asp437 in AnCE, respectively) (Figure 7J).

It is likely that the dipeptides would bind differently/with different affinities to nACE when compared to AnCE due to the differences within the non-prime and prime subsites. The Gln266 (AnCE) substitution to Ser260 (nACE) within the S2′ subsite could affect the stability of tryptophan binding within the subsite. The same substitution would appear to be more favourable for the WR-2 dipeptide, as well as AnCE-Ser268, Phe363, and Asp437 being substituted to nACE- Glu262, Ser357, and Glu432, respectively.

### 3.10. Comparison of Dipeptide Affinity to AnCE, cACE, and nACE

Out of all the dipeptides, IW has the highest affinity for AnCE, with a *K*_i_ value of 80 µM (0.08 mM); this is most likely due to the AnCE subsites being relatively hydrophobic, with any polar interactions stabilising the dipeptide backbone. This dipeptide has also shown to have an increased affinity for cACE compared to nACE, with the *K*_i_ values being 1.6 µM and 59.6 µM, respectively. The VW dipeptide has a slightly lower affinity for AnCE, with a *K*_i_ value of 117 µM; interestingly, VW also has a lower affinity for cACE when compared to IW, with a *K*_i_ value of 2.1 µM. Though VW has lower affinity for cACE and AnCE than IW, it shows higher selectivity when the affinities are compared to nACE, with VW having a *K*_i_ value of 151.6 µM compared to IW having a *K*_i_ value of 59.6 µM. The difference in affinity between IW and VW in AnCE is likely due to the extra carbon atom in isoleucine, which has an effect on the van der Waals interactions within both subsites, potentially further stabilising the dipeptide compared to VW. The differences between cACE and nACE are likely influenced by more than an extra carbon atom. The differences within the subsites are more significant (as discussed above), with one of the key differences that have been shown to influence affinities between nACE and cACE being within the S2′ subsite nACE-Thr358 to cACE-Val380 [38], this substitution can cause the extra carbon to contribute more significantly to ligand affinities.

The YW dipeptide shows low affinity for AnCE, with a *K*_i_ value of 750 µM (0.75 mM). The study by Lunow et al. showed that this dipeptide also has an almost equal affinity for cACE and nACE, with only a slight preference for cACE [14]. The less clear electron density for YW bound to AnCE could be the result of low affinity binding causing the dipeptide to have low occupancy, or it could also be an indication of the tyrosine not being completely stabilised in one conformation, potentially due to the repelling effect that Thr364 could have on the aromatic ring of the tyrosine. This effect would also be present in nACE but not in cACE, which has a valine instead.

Interestingly, RW has a *K*_i_ value of 200 µM (0.2 mM), showing a significantly tighter binding when compared to WR (840 µM). This could be driven by the carboxyl group within the S2′ pocket showing highly favourable interactions with Gln265, His337, Lys495, His497, and Tyr504; the carboxyl interaction with those residues is conserved in all dipeptide structures. This causes unfavourable binding of the side chains of WR, as the ligand positioning is flipped when compared to RW, causing the tryptophan to bind in a more polar environment, resulting in weaker binding of WR in AnCE; this shows how the subsites can accommodate a range of side chains, showing how the promiscuity of these enzymes is achieved. The study carried out by Lunow et al. [14] also shows that RW is a tighter binder in cACE than nACE, but the complete opposite is true for WR, potentially due to the S2′ subsite substitution of cACE-Ser284 and Asp453 to nACE-Glu262 and Glu431. The glutamic acids could favour the WR-2 orientation of binding with arginine towards the S2′ subsite, which is predicted to be maintained in nACE and cACE due to the carboxyl group interaction that appears to be conserved across these enzymes. AnCE and cACE have higher sequence similarity compared to AnCE and nACE, which leads to no surprise that the AnCE data show a similar result to the study by Lunow et al. [14], where the RW ligand has a higher affinity for AnCE compared to WR.

Out of all the dipeptides tested in this study, WR shows the highest *K*_i_ value for AnCE 840 µM (0.84 mM); this partially matches the findings of the study by Lunow et al. [14], which showed that WR has a significantly higher specificity for nACE than cACE. Potentially, the more polar environment created by the AnCE- Ser268, Phe375, Thr387, and Asp437 residues being substituted to nACE- Glu262, Tyr369, Arg381, and Glu431 in the S2 and S2′ pockets are more favourable for the arginine to be at the C-terminal end at the S2 and S2′ subsites.

## 4. Conclusions

The high-resolution crystal structures of AnCE in complex with the dipeptides VW, IW, YW, RW, and WR presented here reveal dual binding in both the non-prime (dipeptide-1) and prime (dipeptide-2) subsites of AnCE. Almost all the dipeptides coordinate the zinc ion through a water-mediated interaction via the carboxyl group for dipeptide-1 and the amine group for dipeptide-2. The only exception to this rule is the WR-1 dipeptide in conformation 2; in this conformation, the amine group is orientated towards the zinc ion, resulting in too long of a distance for coordination.

Kinetic analysis by a fluorometric assay indicated that the dipeptides display micromolar inhibition of AnCE, with the more hydrophobic dipeptides IW and VW displaying the highest affinity. Binding within the prime subsites was shown to be largely influenced by the carboxyl group interacting with the polar pocket created by Gln265, His337, Lys495, His497, and Tyr504. This polar pocket is conserved across all homologous of AnCE and has been shown to stabilise the C-terminus of several inhibitors and substrates of AnCE, nACE, and cACE. In order for the carboxyl group to interact with the polar pocket, the WR-2 orientation in AnCE positions both the tryptophan and arginine in less favourable environments (S1′ and S2′ subsite, respectively), as the S1′ subsite is more polar and the S2′ subsite is more hydrophobic, whereas the RW-2 dipeptide binds in a more favourable orientation (the same as IW, VW, and YW) with the arginine within the S1′ subsite and tryptophan in the S2′ subsite. The favourability in this orientation of binding is also shown by RW having a higher affinity for AnCE than WR (200 µM and 840 µM, respectively). This indicates that the backbone binding within the prime subsite is of greater importance than side chain interactions. This could be greatly due to the nature of AnCE and ACE alike, as these enzymes havedipeptidyl carboxypeptidase activity, and for catalysis to happen at the C-terminal end of ligands, the enzymes must be able to accommodate a wide range of different residues.

These observations also agree with the findings of Lunow et al. [14], which showed that the more hydrophobic dipeptides have a higher affinity to cACE (compared to nACE), whereas YW and WR are poor inhibitors for cACE (Lunow et al., [14]), and the exact same pattern is observed here for AnCE. Based on a comparison of the AnCE crystal structures presented here to the apo structure of nACE and cACE, it can be concluded that these dipeptides would likely bind similarly to cACE, with some differences in binding when compared to nACE, explaining why they show high preferences for individual domains (IW, VW, and RW for cACE and WR for nACE). As this study is limited to AnCE crystal structures, in the future, it would be of benefit to co-crystallise the dipeptides with cACE and nACE.

## Figures and Tables

**Figure 1 biomolecules-15-00591-f001:**
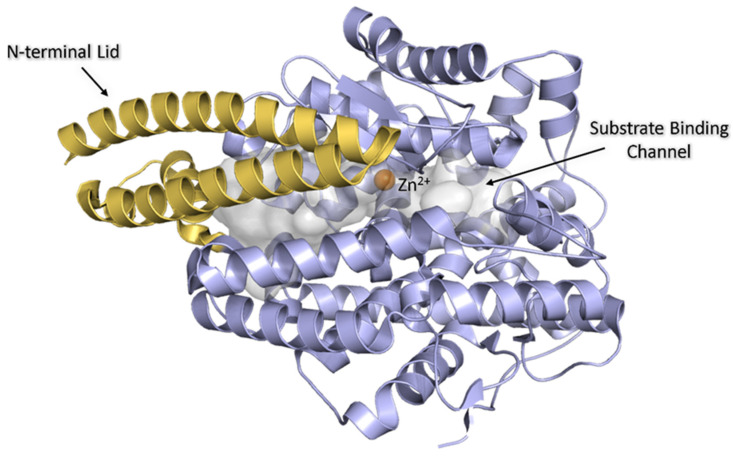
Native AnCE showing lid, zinc ion, and catalytic channel. AnCE is shown in light blue, with the lid represented in yellow orange. The zinc ion is shown in orange inside the catalytic channel displayed in transparent grey. PDB: 5A2R.

**Figure 2 biomolecules-15-00591-f002:**
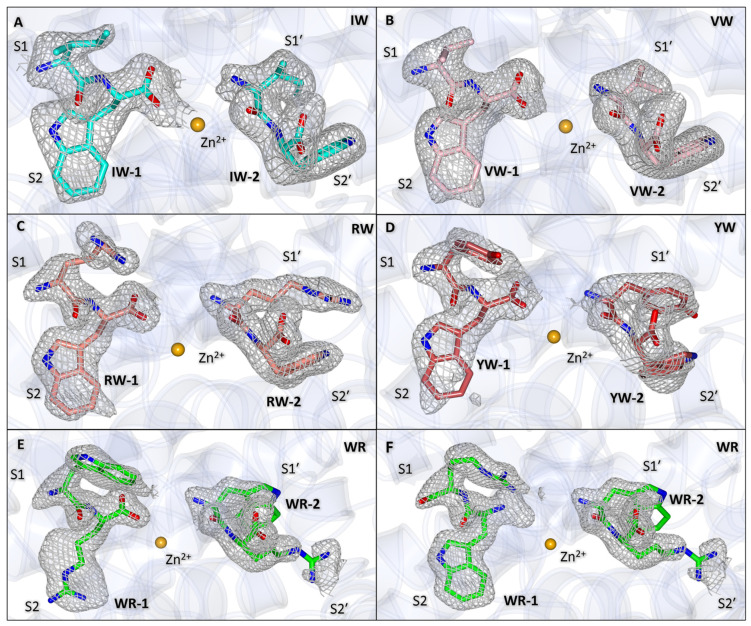
Dipeptide electron density maps. (**A**) IW dipeptide shown in cyan. (**B**) VW shown in light pink. (**C**) RW shown in salmon. (**D**) YW shown in raspberry. (**E**) WR shown in lime, with the non-prime binding in the first conformation. (**F**) WR shown in lime with the non-prime binding in the second conformation. Zinc ion is shown in orange in all panels.

**Figure 3 biomolecules-15-00591-f003:**
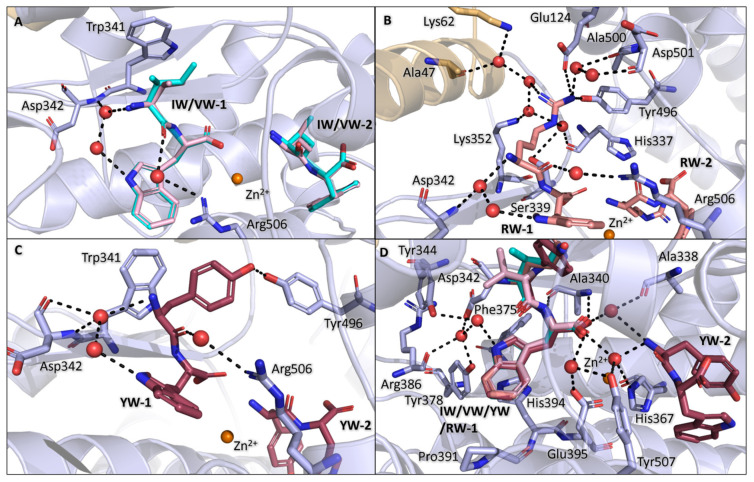
Binding within non-prime subsite. (**A**) IW-1 and VW-1 isoleucine interactions within the subsite; IW is shown in cyan and VW shown in light pink. (**B**) RW-1 arginine interactions within the S1 subsite; RW is shown in salmon. (**C**) YW-1 tyrosine interactions within the subsite; YW is shown in the colour raspberry. (**D**) The tryptophan of IW, VW, YW, and RW are superimposed, showing the same conformation in the S2 subsite and the conserved interactions for all four ligands. In all panels, zinc ion is shown in orange and AnCE residues are in light blue.

**Figure 4 biomolecules-15-00591-f004:**
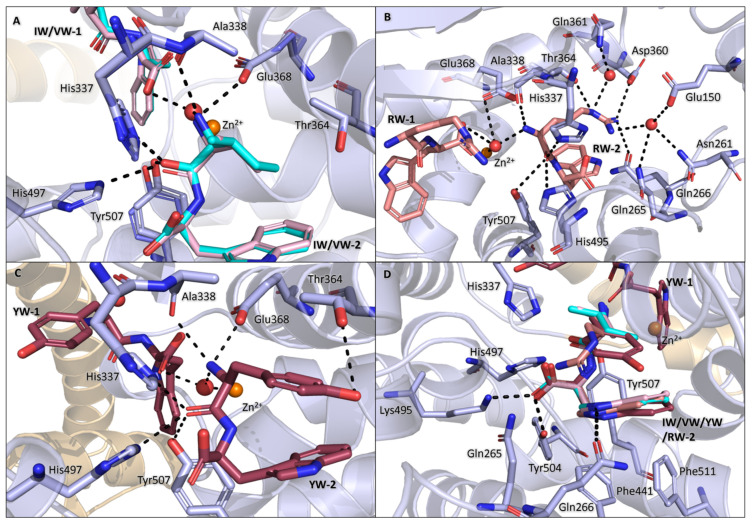
Binding within prime subsite. (**A**) IW-2 and VW-2 are superimposed to show the conserved interactions of isoleucine and valine within the S1′ subsite. IW is shown in cyan, and VW is shown in light pink. (**B**) RW-2 arginine interactions within the S1′ subsite; RW is shown in the colour salmon. (**C**) YW-2 tyrosine interactions within the subsite; YW is shown in raspberry. (**D**) The tryptophan of IW, VW, YW, and RW are superimposed to show the conserved interactions within the S2′ subsite. In all panels, the zinc ion is in the colour orange and the AnCE residues are displayed in light blue.

**Figure 5 biomolecules-15-00591-f005:**
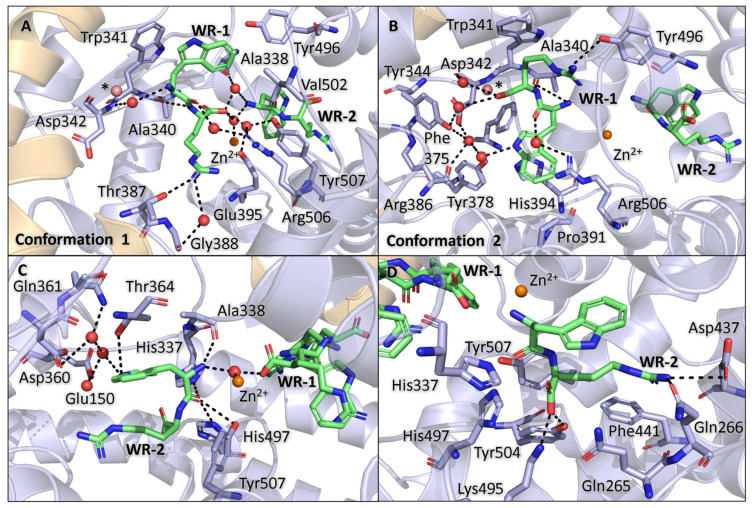
WR interactions with AnCE subsites. (**A**) WR-1 in the first conformation, showing interactions within the S1 and S2 subsites. (**B**) WR-1 in the second conformation, showing interactions with both of the non-prime subsites. (**C**) WR-2 tryptophan interactions within the S1′ subsite. (**D**) WR-2 arginine interactions with the S2′ subsite residues. The dipeptide in all panels is shown in the colour lime. * Water molecule moves depending on WR-1 conformation, the water coloured in dark red is the water present with the shown conformation.

**Figure 6 biomolecules-15-00591-f006:**
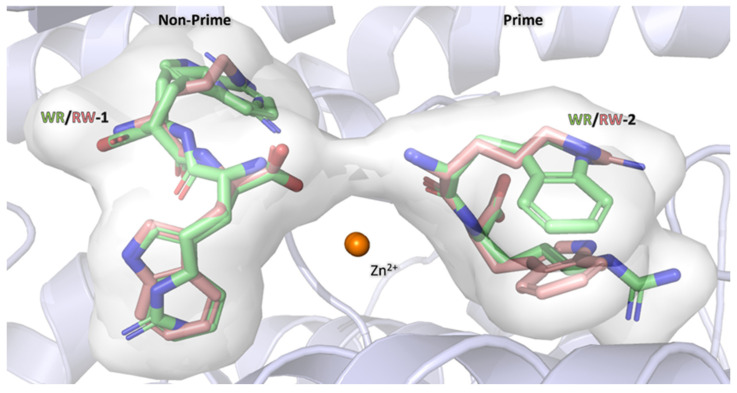
Superimposition of RW and WR showing ligand orientation in space relative to each other. RW is shown in salmon, and WR is represented in lime. The superimposition shows the differences in ligand binding within the non-prime and prime subsites. Within the prime subsite, the backbone position of both ligands is fully conserved.

**Figure 7 biomolecules-15-00591-f007:**
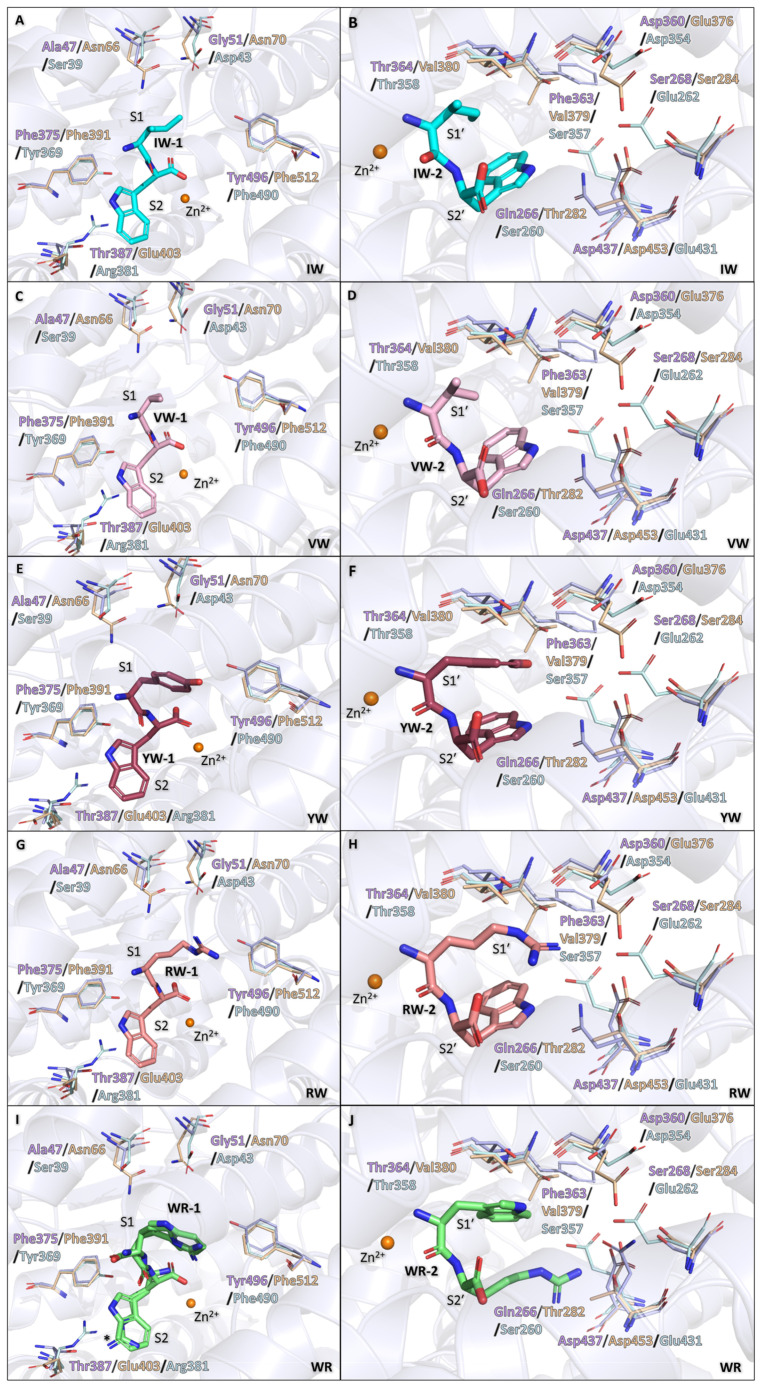
Dipeptides modelled into cACE and nACE active site. (**A**) IW binding within non-prime site. (**B**) IW binding within the prime subsites. (**C**) VW binding within non-prime subsites. (**D**) VW binding within prime subsites. (**E**) YW binding within non-prime subsites. (**F**) YW binding within prime subsites. (**G**) RW binding within non-prime subsites. (**H**) RW binding within prime subsites. (**I**) WR binding within non-prime subsites. (**J**) WR binding within prime subsites. nACE is shown in pale cyan, cACE is shown in the colour wheat, and finally, AnCE is in light blue. * WR-1 shown in dual conformation.

**Figure 8 biomolecules-15-00591-f008:**
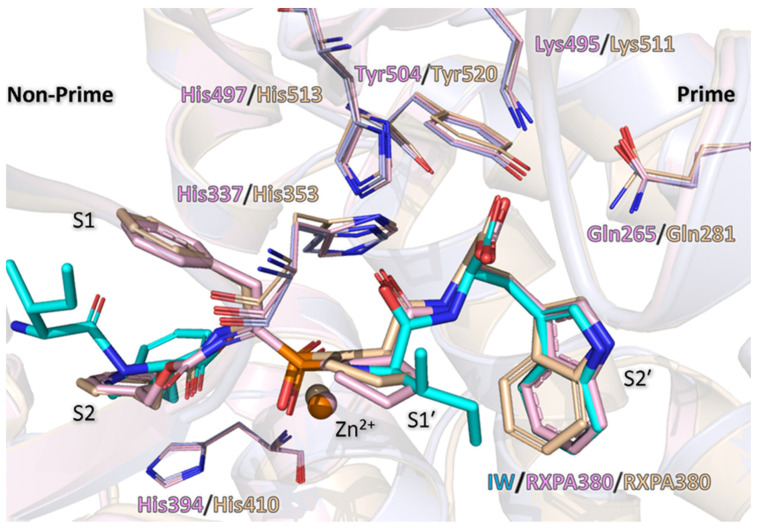
cACE and AnCE superimposed, showing dipeptide and RXP380 binding within the active site. AnCE with dipeptide is shown in light blue, and the dipeptide is shown in cyan; AnCE with RXP380 is shown in light pink (PDB: 2X96); and cACE is shown in wheat colour (PDB:2OC2). Within the non-prime subsites, the dipeptide and RXP380 do not align well; however, the dipeptide tryptophan and RXP380 N-terminal phenylalanine overlap to form an interaction with His394/410, respectively. The prime subsite binding appears to be conserved; this is speculated to be due to the polar pocket interacting with ligand’s carboxyl group.

**Table 1 biomolecules-15-00591-t001:** Dipeptide IC_50_ and *K*_i_ values for AnCE.

Dipeptide	IC_50_ (mM)	*K*_i_ (mM)
IW	0.34	0.08
VW	0.78	0.17
YW	3.40	0.75
RW	0.88	0.20
WR	3.78	0.84

**Table 2 biomolecules-15-00591-t002:** X-ray diffraction data collection, processing, and refinement statistics for AnCE with IW, VW, YW, and WR.

	AnCE_IW	AnCE_VW	AnCE_YW	AnCE_RW	AnCE_WR
Space group	*H3*	*H3*	*H3*	*H3*	*H3*
Unit cell parameters:					
a (Å)	173.2	172.5	172.4	172.9	172.9
b (Å)	173.2	172.5	172.4	172.9	172.9
c (Å)	103.1	103.7	103.5	103.6	103.0
α (o)	90.00	90.00	90.00	90.00	90.0
β (o)	90.00	90.00	90.00	90.00	90.0
γ (o)	120.00	120.00	120.00	120.00	120.0
Molecules per asymmetric unit	1	1	1	1	1
Resolution range (Å)	50.01–2.20 (2.26–2.20)	43.13–2.20 (2.26–2.20)	86.22–2.20 (2.26–2.20)	86.45–1.90 (1.93–1.90)	86.47–1.85 (1.88–1.85)
R_merge_ (%)	19.2 (177.6)	31.9 (383.0)	23.5 (213.0)	12.3 (142.5)	7.9 (90.6)
R_pim_ (%)	6.2 (57.2)	7.0 (84.8)	7.7 (69.2)	4.0 (46.4)	2.6 (32)
Mean *I*/*σ* (*I*)	8.3 (1.3)	8.6 (1.4)	7.0 (1.8)	10.2 (1.5)	15.6 (2.2)
Completeness (%)	100.0 (100.0)	100.0 (99.8)	100.0 (100.0)	100.0 (100.0)	100.0 (100.0)
Number of reflections:					
Total	619,139	1,266,516	596,397	959,889	1,014,896
Unique	58,565	58,412	58,227	91,012	98,093
CC 1/2	0.997 (0.704)	0.997 (0.824)	0.994 (0.599)	0.998 (0.793)	0.999 (0.886)
Multiplicity	10.6 (10.6)	21.7 (21.1)	10.2 (10.2)	10.5 (10.4)	10.3 (8.9)
Average B factor (Å^2^)					
Protein	40.50	40.46	35.94	32.73	30.26
Ligand	91.75	81.13	66.93	62.16	61.84
Zinc ion	34.90	32.61	28.95	28.15	25.55
Solvent	38.20	39.46	36.24	40.21	41.54
Dipeptide-1	57.41	58.93	70.91	54.71	44.16
Dipeptide-2	33.76	44.15	48.91	40.01	54.09
R_work_/R_free_ (%)	0.17/0.21	0.17/0.21	0.16/0.21	0.16/0.20	0.15/0.18
R.M.S deviation from ideal values					
Bond lengths (Å)	0.014	0.008	0.015	0.010	0.011
Bond angles (o)	2.463	1.729	2.560	1.904	1.909
Ramachandran plot statistics (%)					
Favoured	97.48	98.32	98.32	98.83	98.83
Allowed	2.52	1.51	1.68	1.00	1.17
Disallowed	0.00	0.17	0.00	0.17	0.00
Number of non-hydrogen atoms					
Amino acids	4923	4908	4927	4929	4955
Ions	1	1	1	1	1
Ligand	113	100	100	100	137
Water	236	243	309	520	769
Dipeptide-1	23	22	27	26	52
Dipeptide-2	23	22	27	26	26
PDB code	9QA0	9QA1	9QA3	9QA2	9QA4

**Table 3 biomolecules-15-00591-t003:** Active site subsite differences between AnCE, cACE, and nACE.

Subsite	AnCE	cACE	nACE
**S1**	Ala47	Asn66	Ser39
	Gly51	Asn70	Asp43
	Tyr496	Phe512	Phe490
**S2**	Phe375	Phe391	Tyr369
	Thr387	Glu403	Arg381
**S1′**	Asp360	Glu376	Asp354
	Phe363	Val379	Ser357
	Thr364	Val380	Thr358
**S2′**	Gln266	Thr282	Ser260
	Ser268	Ser284	Glu262
	Asp437	Asp453	Glu431

## Data Availability

The atomic coordinates and structure factors for the AnCE_IW, AnCE_VW, AnCE_YW, AnCE_RW, and AnCE_WR complexes have been deposited with the Protein Data Bank under accession codes 9QA0, 9QA1, 9QA3, 9QA2, and 9QA4, respectively.

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
