# Peer review of "Molecular Basis of Dipeptide Recognition in Drosophila melanogaster Angiotensin I-Converting Enzyme Homologue, AnCE"

_biomolecules, 2025, doi:10.3390/biom15040591_

Round 1

Reviewer 1 Report

Comments and Suggestions for Authors

In this paper, enzyme kinetics experiments and X-ray crystallography techniques are used to reveal how dipeptides bind AnCE as selective inhibitors, and to explore how these findings can be applied to the C-terminal and n-terminal domains of human angiotensin I converting enzyme (ACE) to develop domain-specific ACE inhibitors. The objectives and the rationale of the study are clearly stated, however the paper still needs improvement before acceptance for publication. My detailed comments are as follows:

  1. The ultimate purpose of this paper is to guide the development of domain-specific ACE inhibitors. Why not just use the C-and N-terminals of ACE instead of AnCE?Is it just that AnCE is easier to crystallize?
  2. How to obtain the binding patterns of the C-and N-terminals of ACE from the eutectic structure information of dipeptide and AnCE? The article needs a supplementary description.
  3. Were repeated experiments carried out in kineticexperiments of AnCE inhibition? Errors should be added to the data in Table 1.

Author Response

Reviewer 1

  1. The ultimate purpose of this paper is to guide the development of domain-specific ACE inhibitors. Why not just use the C-and N-terminals of ACE instead of AnCE? Is it just that AnCE is easier to crystallize?

AnCE can be easily expressed and purified with a final yield of 20 mg from 1 L of culture. To date as successful of an expressions and purifications has not been achieved for ACE. AnCE also crystallises much easier compared to cACE or nACE. These points have been addressed in the manuscript introduction lines 85-87.

  1. How to obtain the binding patterns of the C-and N-terminals of ACE from the eutectic structure information of dipeptide and AnCE? The article needs a supplementary description. 

Potentially this comment was misunderstood, but cACE and nACE modelling of the dipeptides performed in this manuscript was done through superimposition of the AnCE dipeptide bound structures onto the apo structures of cACE and nCAE. This is described in the manuscript in lines 438-439 and 516-517.

  1. Were repeated experiments carried out in kinetic experiments of AnCE inhibition? Errors should be added to the data in Table 1. 

The methods for the kinetic assay have been updated to list the R2 values. This can be found in the manuscript lines 162-176.

Reviewer 2 Report

Comments and Suggestions for Authors

Joanna Å»ukowska and collaborators presented an article titled, “Molecular basis of dipeptide recognition in Drosophila melanogaster angiotensin I-converting enzyme homolog, AnCE”. This research aims to conduct enzyme kinetic assays and X-ray crystallography

techniques and to the possible use of dipeptides as selective inhibitors for AnCE. The authors determined the binding of all dipeptides to AnCE in two distinct locations, the non-prime and prime subsites and identified the significant role of hydrophobic and aromatic amino acid residues. The authors also demonstrated that the S2′ subsite has a major influence on the binding orientation within the prime subsites and the possibility of utilizing the dipeptides as potential selective inhibitors of the enzyme ACE.

The introduction section covered the relevant literature highlighting the protein enzyme targets and their structural features and functions including the currently available inhibitors. The authors focused primarily on dipeptides of RW, WR, VW, IW, and YW in the present study and co-crystallized with AnCE, as suitable models to study the functions and role of ACE inhibition. The authors also referred to their previous research publications to support the aim of the work. The concept is well designed and the manuscript is presented in a lucid way. This study indicated the ease of crystallization and purification strategies which are significantly challenging to the researchers and the study added advantages to the existing knowledge. The authors followed standard protocols for the conducted experiments involving AnCE expression, purification, crystallization, X-ray diffraction data collection, and structure determination, followed by inhibition of AnCE activity. The kinetic analysis results are quite exciting with the studied 5 peptides and all the validation parameters confirmed their corresponding structures. All the data collected in tables and all figures are presented clearly in the manuscript. All the co-crystallized structural data was deposited to PDB as per the guidelines. Two interesting peptides showed exceptionally better selectivity and binding affinity. The results of binding within the prime subsites and non-prime subsites were discussed in comparison to them, extensively. The interesting point is the superimposition of RW and WR showing ligand orientation in space relative to each other in consideration of their relative conformation. The manuscript is suitable for publication and the following minor comments need to be addressed before the acceptance.

  1. In the introduction, some recent literature needs to be discussed relevant to the study.
  2. Authors may include the limitations and future scope of the present study in the conclusion section.
  3. The abbreviation section can be deleted, and these 5 abbreviations can be included inside the manuscript, where they appear first in the text.
  4. Wherever possible, the instrument manufacturer's name, city, and country must be mentioned, accordingly.

Author Response

Reviewer 2

  1. In the introduction, some recent literature needs to be discussed relevant to the study.

A couple more recent references were included in the introduction (lines 47 and 58).

  1. Authors may include the limitations and future scope of the present study in the conclusion section. 

A sentence has been added to the conclusion about future work (lines 658-659). 

  1. The abbreviation section can be deleted, and these 5 abbreviations can be included inside the manuscript, where they appear first in the text.

Abbreviations were included in accordance with the journal template guidelines. They are also included within the manuscript when they first appear in text.

  1. Wherever possible, the instrument manufacturer's name, city, and country must be mentioned, accordingly.

Instruments manufacturer city and country was included on lines 111-112 and 116.

Reviewer 3 Report

Comments and Suggestions for Authors

The human ACE (Angiotensin converting enzyme) plays a key role in blood pressure and volume control, is an important factor in chronic inflammation and fibrotization processes in cardiovascular tissues; several millions of patients are taking regularly ACE inhibitor drugs. Revealing the molecular dynamics in the active center during enzyme action and inhibition thus has great theoretical and medical importance, study of nonvertebral forms of the enzyme can give useful hints for understanding the working of the human molecule. The submitted paper is one in a series of publications in which the best experts of the field are analyzing steric effects that happen in the active center of this enzyme during interaction with different characteristic dipeptides and inhibitor molecules. Enzyme binding, enzymatic activity and X-ray diffraction studies have been performed to determine the steps of substrate binding and leading to the breaking of the peptide bond for the ACE enzyme of Drosophila melanogaster. 
The statement that nACE  and cACE  domains might have separate functions should be taken with more care. Ang II produced by both has both hypertensive and chronic inflammatory effects, but differences can be present in other functions.
The paper is well built, excellently organized, style, logical reasoning, Tables, Figures reflect high level of professionalism.
An X-ray diffraction figure based on which the atomic distances have been computed should be demonstrated. 
The AnCE has been produced in transgenic yeast cells. Also, it is mentioned that its crystallization properties are better than those of human cACE. One can not exclude the possibility, that slight differences in amino acid sequence, or posttranslational modifications (glycosylation) of the secreted protein might induce topological changes  affecting the area of the active center, too. A recent paper by the lab dealing directly with the  human somatic ACE might yield some valuable comparison (Gregory KS et al FEBS Journal 2025;292:1141). This paper revealed that amino acid sequence differences farther from the active center can play a role in substrate or antagonist binding specificity. 

Author Response

Reviewer 3

  1. The statement that nACE and cACE domains might have separate functions should be taken with more care. Ang II produced by both has both hypertensive and chronic inflammatory effects, but differences can be present in other functions.

On line 57 a small edit was made to help distinguish that though the domains have dominant functions, they do also have other functions.

  1. An X-ray diffraction figure based on which the atomic distances have been computed should be demonstrated.

Unsure what is meant by this comment. Diffractions images are not normally included in publications.

  1. The AnCE has been produced in transgenic yeast cells. Also, it is mentioned that its crystallization properties are better than those of human cACE. One cannot exclude the possibility, that slight differences in amino acid sequence, or posttranslational modifications (glycosylation) of the secreted protein might induce topological changes affecting the area of the active centre, too. A recent paper by the lab dealing directly with the human somatic ACE might yield some valuable comparison (Gregory KS et al FEBS Journal 2025;292:1141). This paper revealed that amino acid sequence differences farther from the active centre can play a role in substrate or antagonist binding specificity.

It has been shown that ACE could have distant residues involved in substrate specificity. However, in this manuscript only the conformation of binding within the active site is speculated for cACE and nACE. To date the distant residues of ACE involved in substrate specificity have not been identified, due to this, a direct comparisons cannot be made between AnCE and ACE binding with respect to those residues.

Also, comparisons of AnCE structure with cACE and nACE structures have shown no differences in topology.   

Round 2

Reviewer 1 Report

Comments and Suggestions for Authors

The author has revised the manuscript as required, thus it could be accept in present form.

Reviewer 2 Report

Comments and Suggestions for Authors

The authors have revised substantially hence, the manuscript can be accepted in its present form.

Reviewer 3 Report

Comments and Suggestions for Authors

Description of enzyme binding and inhibition tests have been added.  Some minor corrections, have been performed.